# Fetal Tachyarrhythmia Management from Digoxin to Amiodarone—A Review

**DOI:** 10.3390/jcm11030804

**Published:** 2022-02-02

**Authors:** Liliana Gozar, Dorottya Gabor-Miklosi, Rodica Toganel, Amalia Fagarasan, Horea Gozar, Daniela Toma, Andreea Cerghit-Paler

**Affiliations:** 1Department of Pediatrics, “George Emil Palade” University of Medicine, Pharmacy, Science, and Technology of Târgu-Mureș, 540139 Târgu-Mureș, Romania; liliana.gozar@umfst.ro (L.G.); rodica.toganel@umfst.ro (R.T.); amalia.fagarasan@umfst.ro (A.F.); daniela.toma@umfst.ro (D.T.); palerandreea@yahoo.com (A.C.-P.); 2Pediatric Cardiology, Emergency Institute of Cardiovascular Diseases and Transplantation, 540139 Târgu-Mureș, Romania; 3Department of Pediatric Surgery, “George Emil Palade” University of Medicine, Pharmacy, Science, and Technology of Târgu-Mureș, 540139 Târgu-Mureș, Romania; horea.gozar@umfst.ro

**Keywords:** arrhythmia, fetal, transplacental medication, management, treatment protocol

## Abstract

Sustained fetal tachycardias are rare but represent a high risk of mortality and morbidity. Consensus has yet to be found regarding their optimal management. The aim of this narrative review is to summarize the data available in the current literature regarding the efficacy and safety of medications used in the management of intrauterine tachyarrhythmias and to provide possible treatment protocols. In this review, we would like to emphasize the importance of a thorough evaluation of both the fetus and the mother, prior to transplacental antiarrhythmic drug initiation. Factors such as the hemodynamic status of the fetus, possible mechanisms of fetal arrhythmia, and concomitant maternal conditions are of primordial importance. As a possible treatment protocol, we would like to recommend the following: due to the risk of sustained supraventricular tachycardia (SVT), fetuses with frequent premature atrial beats should be evaluated more frequently by echocardiography. A careful hemodynamic evaluation of a fetus with tachycardia is primordial in forestalling the appearance of hydrops. In the case of atrial flutter (AFL), sotalol therapy could represent a first choice, whereas when dealing with SVT patients, flecainide should be considered, especially for hydropic patients. These data require consolidation through larger scale, non-randomized studies and should be handled with caution.

## 1. Introduction

Fetal arrhythmias can occur in approximately 1% of pregnancies. The majority of these cardiac rhythm disorders are represented by benign conditions, such as premature atrial contractions, that do not require treatment [1]. Sustained fetal tachycardias are rare, occurring only in up to 1 of 1000 pregnancies [1], but it is well known that these dysrhythmias may lead to hemodynamic decompensation, hydrops, premature birth, or even perinatal death [2]. Behind such an unfavorable evolution may lie the lack of an early diagnosis, before the appearance of fetal hydrops.

The most commonly diagnosed arrhythmias are accessory pathway-mediated supraventricular tachycardias (SVTs), followed by atrial flutter (AFL), and ectopic atrial tachycardia [3,4]. AFL can also occur secondary to myocarditis, structural congenital heart disease, or SSA autoantibodies [5].On the other hand, atrial fibrillation [6] and junctional ectopic tachycardia represent a rarity in this age group [7]. The latter is most often associated with the presence of SSA/Ro antibodies, with or without atrioventricular (AV) block [8,9]. In such situations, the therapy must be directed towards its cause.

The assessment of these rhythm disturbances may represent a challenge, echocardiography serving as an important, wide spread and available diagnostic tool of evaluation, through M-mode and Doppler techniques. Establishing the relationship between atrial and ventricular activity and the measurement of the ventriculoatrial (VA) interval is key to the diagnosis of the tachycardia [10,11,12]. Another diagnostic possibility is represented by the tissue Doppler method, allowing the diagnosis of the arrhythmia through a temporary recording of atrial and ventricular activities [13]. Contrary to telemetry, echocardiography (which remains the method of choice in the diagnostic process) does not allow a prolonged monitoring of the fetal rhythm [13]. This deficiency may be one of the reasons for the progression of an undiagnosed tachyarrhythmia and the appearance of hydrops [14].

The determination of the underlying arrhythmia mechanism holds great prognostic value and may represent an important aspect in choosing the appropriate transplacental medication, when needed [14].

Consensus has yet to be found regarding the optimal treatment of fetal tachycardias. In this setting, the aim of this narrative review is to summarize the data available in the current literature regarding the efficacy and safety of the antiarrhythmic medications used in the management of intrauterine tachyarrhythmias and to provide possible treatment protocols.

## 2. Materials and Methods

A comprehensive search was conducted in February 2021, using the PubMed database and the search engine Google Scholar, in conformity with methods provided by the current scientific guidelines [15]. We reviewed data from the literature, from the first reports published regarding the management of fetal tachycardia to the most recent studies available. A manual search of the reference lists of all retrieved articles was also conducted. The search was limited to English language publications and was repeated prior to submission to locate any recent, additional publications.

Three independent reviewers were included in the whole review process, which consisted firstly of a screening for titles and abstracts in conformity with both the inclusion and the exclusion criteria. After the removal of irrelevant studies or duplicates, a full-text screening was performed to assess eligibility for final inclusion. Inter-rater reliability was also conducted. As a limitation, we would like to mention that no review software was used in the process.

The main outcomes sought were termination of the tachycardia, termination rates, time of conversion to sinus rhythm, modifications in the hemodynamic status of the fetus, fetal death, and medication associated with fetal or maternal complications.

The search terms included the following: “fetal”, “fetus”, “tachyarrhythmia”, “tachycardia”, “treatment”, “management”, “arrhythmia”, “antiarrhythmic”.

Only original studies conducted on a significant number of patients (>20) were included. Thus, case presentations and small studies were excluded to help improve the accuracy of the results. Categorization of the cases based on their probable arrhythmia mechanism represented an important inclusion criterion.

Data extracted by a single author consisted of the number of total patients, the number of patients with hydrops, gestational age, type of tachycardia, applied treatment protocol, drug dosage, termination rates, and medication side effects.

## 3. Results

The studies included in our review covered a large period, from 1980 to 2020; thus, great developments in the field of fetal cardiology could be observed through their review. Fifteen studies met the inclusion criteria, with 839 patients. The results are presented in Table 1, Table 2 and Table 3. Antiarrhythmic medications and their recommended dosages are outlined in Table 1, whereas Table 2 and Table 3 present a short summary of the included studies.

## 4. Discussion

Due to its relatively safe profile, long history of use in pregnancies, and the familiarity with its use in many centers, digoxin represents the first-choice drug when dealing with sustained fetal tachyarrhythmias [23]. Among the first studies sharing the experience regarding a large group of fetuses receiving antiarrhythmic medication, are those published by Kleinmann (1985) [32] and Maxwell (1988) [33]. The aforementioned studies published results regarding digoxin monotherapy or a combination of digoxin with propranolol, procainamide, and verapamil, but failed to confirm their efficacy [32,33]. Nevertheless, digoxin is still maintained in therapeutic protocols. Due to its effect on AV conduction, it can cause a decrease in the ventricular rate and it can expose an apparently regular, long VA tachycardia.

### 4.1. Digoxin versus Flecainide in Fetal Therapy

Several studies have been published in the literature that address the topic of fetal tachycardia management, but the results of only a few of them are based on a significant number of cases. Regardless of its retrospective nature, the study conducted by Simpson et al. is considered to be of great importance, as it shares important insight about conversion rates in different settings [16]. The study group consisted of a total of 127 fetuses, of which 105 were diagnosed with SVT, and the remaining 22 with AFL; 52 fetuses were hydropic. The drugs included in the study were digoxin, verapamil, and flecainide. The conversion rate in non-hydropic patients receiving digoxin monotherapy was of 63% [16].

In 2019, the Japanese Fetal Arrhythmia Group published the results of a prospective single-arm multicenter trial, where 50 patients from 15 Japanese institutions where followed [23]. The aim of the aforementioned study was to evaluate the efficacy of a protocol defined transplacental treatment for fetal SVT and AFL [23]. For fetuses with short VA SVT, or AFL, but without hydrops, the first-choice drug was digoxin [23,34]. Digoxin as a monotherapeutic first-line drug had a resolution rate of 47% in non-hydropic, short-VA SVT patients and a 59% resolution rate in AFL patients [23].

In a retrospective study comparing two drug protocols (with either flecainide or digoxin used as a first-line option in two tertiary institutions), 38% (19/50) of the patients receiving digoxin required the association of a second drug, whilst in the flecainide group, this rate was as low as 9% (3/34) [21]. A higher demand for a therapy change in the case of administration of digoxin as a monotherapeutic first-line agent was published by Strizek et al., who reported that in 8/10 (80%) of the examined cases, digoxin failed to convert the arrhythmia, and its substitution or association with another agent (flecainide) was necessary [28].

The aim of a meta-analysis conducted by Alsaied et al. was to compare first-line transplacentar antiarrhythmics used in monotherapy for fetal SVTs [1]. The results of the 10 studied included proved flecainide to be more effective than digoxin, when comparing its termination rate of fetal SVTs to that of digoxin in both hydropic and non-hydropic patients [1]. Regarding the incidence of maternal side effects and fetal death, there were no differences between the two medications [1].

### 4.2. Flecainide and Sotalol inFetal Therapy

The study published by Simpson highlighted that after the addition of verapamil or the substitution of digoxin with flecainide, the resolution rate was 83% [16]. The overall conversion rate in hydropic patients was lower (63%) when compared to that in their non-hydropic counterparts [16].

Since then, some concerns have been raised regarding the efficacy of digoxin, with several authors advocating for the use of flecainide or sotalol as first-line medications [1,19,21,28,29,35].

As a second- and third-line therapy, sotalol and flecainide were given in addition to digoxin [15,35]. In the case of fetuses with short VA SVTs and hydrops, the fetal therapy was started with the second-line therapy (digoxin and sotalol) [23,34]. For fetuses with long VA SVT, sotalol and flecainide were used as first- and second-line treatment [23,34].

Following the above-mentioned protocol, fetal tachyarrhythmia resolved in 89.8% (44/49) of the cases [23]. All long-VA SVT cases (*n* = 4) were effectively treated [23].

When comparing the two main causes of fetal tachyarrhythmia, i.e., AFL and SVT, a multicenter nonrandomized study concluded that AFL was likely to respond more slowly to therapy, and sotalol as a first-line treatment option was associated with a higher termination rate [18]. The study by van der Heijden et al. evaluated the efficacy of sotalol for fetal tachycardias and somewhat confirmed the results [25]. The conversion rates of sotalol as a monotherapeutic first-line agent in fetal atrial flutter were reported to be as high as 80% (8/10) [24,25]. On the other hand, Jaeggi et al. also concluded that both digoxin and flecainide proved more effective than sotalol as regards conversion rates, and tachycardia rates were decreased more effectively in fetal SVT [18].

A Turkish group conducted a retrospective analysis on 23 consecutive patients with fetal tachyarrhythmias [19]. In the aforementioned study group, 17/23 patients required transplacental treatment, and in all of the cases flecainide was used as a first-line medication. The reported success rate for flecainide as a first-line, monotherapeutic agent was as high as 88.2% [19]. Similarly to other reports, the authors concluded that flecainide may be safely used as a first-line treatment for fetal SVTs [1,19,21,28].

Previous concerns were raised regarding the usage of flecainide, due to its high rate of intrauterine death; however, recent studies did not confirm this finding [1,21]. An explication may lie behind the preferential use of flecainide in hydropic fetuses, due to reduced transplacental transport of digoxin in this setting, hydrops being associated with a high mortality rate [1,21]. In a retrospective study, Sridharan et al. concluded that no adverse fetal outcomes were found in the hydropic group receiving flecainide, whilst neonatal or fetal death occurred in 41% (9/21) of hydropic fetuses under digoxin treatment [21].

Similar results were published by Hill et al.: through the analysis of 21 eligible studies, the group found digoxin to be inferior when compared to flecainide and sotalol, the difference being even more evident in hydropic patients [22]. Ekiz et al. concluded in a retrospective study, that in 88.2% (13/15) of hydropic fetuses, a successful conversion to sinus rhythm was achieved with flecainide monotherapy, thus advocating for its efficacy [19].

In the management of intrauterine tachyarrhythmias, the time of conversion to sinus rhythm or the achievement of rate control is of primordial importance. Sridharan et al. found that the conversion median was lower when flecainide was administered (3 days) in comparison to digoxin (8 days) [21]. Similar results were published by a German group, which found no statistical difference regarding the conversion time of flecainide in hydropic versus non-hydropic patients [28].

When observing the efficacy of sotalol, van der Heijden et al. concluded that the conversion rates were lower (sinus rhythm achieved in 4/6, i.e., 67% of the patients) and the time of conversion (median 7.5 days) was longer in hydropic patients when compared to their non-hydropic counterparts (conversion rate 18/22, 81% of the patients; time to conversion, median 3.5 days) [25]. On the other hand, Oudjik et al. reported in a retrospective study, that there were no differences between hydropic and non-hydropic patients regarding the time of successful conversion to sinus rhythm by sotalol therapy, thus confirming its efficacy in hydropic patients as well [24].

The results of the study conducted by Hansmann et al. are debatable and should be handled with caution, as digoxin represented the first-line therapeutic option, even in the severely hydropic patients [2]. Throughout the latest developments in the field of fetal cardiology, digoxin is considered to be inferior to both flecainide and sotalol in fetal hydrops, several authors advocating for their efficacy and usage as first-line recommendation [19,36].

### 4.3. The Place of Amiodarone in Therapy

Due to its significant toxicity profile, amiodarone is recommended only as a third-line therapeutic option, in drug-refractory tachycardias [17,20,27].

In severe hydropic, refractory cases to transplacental therapy, direct fetal treatment was also attempted [2,16,37]. Hansmann et al. in 1991 published the results of their study, conducted between 1981 and 1990. Of the 60 fetuses included, 26 presented signs of hydrops. The treatment protocol involved the transplacental administration of digoxin as a first-line medication, followed by the association with other antiarrhythmics, most frequently verapamil, but propafenone, quinidine, or even amiodarone was also administered [2]. In severe, hydropic, refractory cases, direct fetal therapy was performed (*n* = 13) in addition to the transplacental medication, with amiodarone showing good results in this clinical setting [2]. The survival rate for hydropic fetuses was 20/26 [2].

### 4.4. Maternal Side Effects

Maternal tolerance of antiarrhythmic medications represent an important factor in the management of fetal tachycardias, and the choice of the most efficient drug in the lowest effective dose in this clinical setting is of primordial importance [34].

Measures such as the evaluation and correction of maternal electrolyte disorders (Ca, Mg, K) and vitamin D levels could help improve the response and the conversion to sinus rhythm in transplacental therapy [38].

The study published by Chimenea A. et al. addresses the maternal effects of digoxin, concluding that side effects occur when the serum digoxin level is over 2 ng/mL, but are mild and limited [39].

Combination therapies represent a greater risk for both maternal and fetal adverse events than monotherapies [20].

Some studies reported no significant differences between the incidence of maternal side effects when comparing digoxin, flecainide, and sotalol [1,36]. Reported maternal adverse events mainly consisted of minor symptoms, such as nausea, dizziness, or vomiting [18,23]. Although representing a rare entity, serious events have also been reported: a combination therapy (sotalol and digoxin) induced Mobitz type II AV block [23] or maternal atrial fibrillation, observed after flecainide treatment [19], but with spontaneous resolution of both complications after discontinuation of the therapy. In this setting, the need for a careful maternal monitoring becomes evident.

Malhamé et al. proposed a monitoring protocol for women during the administration of antiarrhythmic therapy for fetal SVT [22]. Maternal cardiac history, physical examination, review of maternal medications (special attention to QTc prolonging agents—antiemetics and antibiotics), baseline ECG, renal and hepatic function, electrolyte levels should be obtained for all patients, and a consultation with an adult cardiologist is also recommended [22].

Due to the lack of clinical guidelines, throughout the literature several treatment protocols are reported (Table 1 and Table 2).

### 4.5. Limitations

The studies included in our review covered a large period, from 1980 to 2020, where fetal and pediatric cardiology underwent a considerable transformation, thus great developments in the field could be observed through our review. The main limitation of this study is that the majority of the included studies were retrospective, and the drug choice was provider-dependent. Moreover, the data for some outcomes such as time of arrhythmia termination, fetal growth restriction, rate of prematurity, neonatal mortality, and maternal side effects were insufficient in some of the studies. Another limitation of the review is the relatively small number of patients included, because of the rarity of fetal tachycardia, the relatively small number of studies addressing the treatment of this pathology, and the small sample sizes in all evaluated studies.

Despite these limitations, we believe that this review provides valid information about fetal tachycardia treatment. Further prospective studies are essential to evaluate the best treatment approach for fetal tachycardia.

## 5. Conclusions

As a conclusion, we would like to emphasize the importance of a thorough evaluation not only of the fetus, but also of the mother, prior to transplacental antiarrhythmic drug initiation. Factors such as the hemodynamic status of the fetus, the presence of hydrops, possible mechanisms of the fetal arrhythmia, and concomitant maternal conditions are of primordial importance and represent important aspects that need to be considered when dealing with fetal arrhythmias.

Direct fetal therapy is reserved for extreme cases, and administration of more than two antiarrhythmics is recommended only in exceptional situations.

As a possible treatment protocol, we would like to recommend the following: due to the risk of sustained SVT, fetuses with frequent premature atrial beats should be evaluated more frequently by echocardiography. A careful hemodynamic evaluation of a fetus with tachycardia is of primordial importance in forestalling the appearance of hydrops. In the case of AFL, we suggest that sotalol therapy is the first-line drug choice, whereas when dealing with SVT, especially in hydropic patients, flecainide is our first-line drug recommendation, based on our review. These data require consolidation through larger scale non-randomized studies and should be handled with caution.

## Figures and Tables

**Table 1 jcm-11-00804-t001:** Transplacental antiarrhythmic drugs and recommended dosages.

Medication	Reference	Dosage
Digoxin	Simpson et al., 1998 [16] Jouannic et al., 2003 [17]	3 × 0.25 mg/day
	Jaeggi et al., 2011 [18] Ekiz et al., 2018 [19]	LD: 1.5–2 mg over 2 days p.o. MD: 0.375–1 mg/day p.o. Aim, maternal drug levels: 2–2.5 ng/mL
	Donofrio et al., 2014 [20]	LD: 1200–1500 µg/24 h i.v., divided every 8 h MD: 375–750 µg/day every 8 to 12 h p.o.
	Sridharan et al., 2016 [21]	LD: 1500 µg/24 h i.v., divided every 8 h, increasing up to 2000 µg/24 h i.v. every 12 h if required Administered until a maternal plasma level of 2.0–3.0 ng/mL was achieved Once therapeutic levels were obtained, administration was switched to p.o.
	Malhamé et al., 2018 [22] Miyoshi et al., 2019 [23]	LD: 0.5 mg i.v. administered once, followed by two doses of 0.25 mg i.v. every 8 h (p.o. 1.5 mg/day divided every 8 h) MD: 0.75 mg/day divided every 8 h p.o. Aim, maternal drug levels: 1.5–2 ng/mL.
Sotalol	Oudjik et al., 2000 [24]	80–160 mg two times a day Maximum dose of 3 × 160 mg/day
	Jaeggi et al., 2011 [18]	160–480 mg/day divided every 12 h p.o. Hydropic fetus starting dose: 320 mg/day
	van der Heijden et al., 2013 [25]	160–320 mg/day divided every 8 to 12 h p.o.
	Donofrio et al., 2014 [20]	160–480 mg/day divided every 8 to 12 h p.o.
	Malhamé et al., 2018 [22]	240 mg/day divided every 8 h p.o.
	Miyoshi et al., 2019 [23]	160–320 mg/day divided every 8 h p.o.
Flecainide	Simpson et al., 1998 [16] Jouannic et al., 2003 [17] Jaeggi et al., 2011 [18] Vigneswaran et al., 2014 [26] Sridharan et al., 2016 [21] Malhamé et al., 2018 [22] Ekiz et al., 2018 [19]	300 mg/day divided every 8 h p.o.
	Donofrio et al., 2014 [20]	100–300 mg/day divided every 8 to 12 h p.o.
	Miyoshi et al., 2019 [23]	200–300 mg/day divided every 12 h p.o.
Amiodarone	Jouannic et al., 2003 [17]	1600–2000 mg/day for two days, then reduced to 400–600 mg/day
	Donofrio et al., 2014 [20]	LD: 1800–2400 mg/d divided every 6 h for 48 h p.o.; lower (800–1200 mg p.o.) if prior frug therapy MD: 200–600 mg/day p.o.

i.v.—intravenous; LD—loading dose; MD—maintenance dose; p.o.—orally; h—hour.

**Table 2 jcm-11-00804-t002:** Summary of study protocols found in the literature.

Authors	Study Period	Design	Drugs	Administration Route	Total No. of Patients	Hydrops (Patient No)	Tachycardia Type (Patient No)				
							SVT Mechanism Not Specified	Long VA	Short VA	AFL	VT
Simpson et al. [16]	1980–1996	R	Digoxin, Verapamil, Flecainide	Direct, transplacental	127	52	105	-	-	22	-
Hansmann et al. [2]	1981–1990	Single-center, prospective	Digoxin, Verapamil, Amiodarone, Propafenone, Quinidine, Flecainide	Direct, transplacental	60	26	54	-	-	6	-
Jouannic et al. [17]	1990–2001	R	Amiodarone	Transplacental	26	26	-	-	22	4	-
Strasbourger et al. [27]	1990–2002	R	Digoxin, Sotalol, Amiodarone	Transplacental	26	24	-	1	15	9	1
Oudjik et al. [24]	1993–1999	R	Sotalol	Transplacental	21	9	10	-	-	10	1
Vigneswaran et al. [26]	1997–2012	R	Flecainide	Transplacental	32	15	-	4	28	-	-
Jaeggi et al. [18]	1998–2008	R	Digoxin, Sotalol, Flecainide	Transplacental	159	35	-	16	98	45	-
Sridharam et al. [21]	1998–2012	R	Flecainide, Digoxin	Transplacental	84	28	-	17	67	-	-
Strizek et al. [28]	2002–2014	R	Flecainide, Digoxin	Transplacental	48	22	-	3	43	2	-
Van Der Heijden et al. [25]	2004–2010	R	Sotalol	Transplacental	28	6	18	-	-	10	
Shah et al. [29]	2004–2008	R	Sotalol	Transplacental	21	8	16	-	-	5	-
Miyoshi et al. [23]	2010–2017	Multicenter, single-arm trial	Digoxin, Sotalol, Flecainide	Transplacental	50	4	-	4	17	29	-
Ekiz et al. [19]	2011–2016	R	Flecainide	Transplacental	23	16	-	1	20	2	-
O’Leary et al. [30]	1985–2018	R	Digoxin, Sotalol, Flecainide, Amiodarone	Transplacental	65	13	-	8	41	16	-
Broom et al. [31]	2000–2020	R	Digoxin, Flecainide, Sotalol	Transplacental	69	23	62	-	-	7	-

VA—ventriculoatrial; AFL—atrial flutter; VT—ventricular tachycardia; SVT—supraventricular tachycardia; R—retrospective.

**Table 3 jcm-11-00804-t003:** Summary of study protocols found in the literature (continuation).

Authors	Conversion Rate	Adverse Events
Simpson et al. [16]	NH: 83% H: 66%	Not reported
Hansmann et al. [2]	Only survival rates reported!	Not reported
Jouannic et al. [17]	Overall: 60%	Fetal: 2/11 postnatal thyroid dysfunction Maternal: 1/26 abdominal pain, vomiting (combination therapy—digoxin and amiodarone)
Strasbourger et al. [27]	Reentry SVT: 14/15 (93%) VT/JET: 2/2 (100%) AFL: 3/9 (33%)	Fetal: 5/26 postnatal thyroid dysfunction Maternal: 1/26 photosensitivity and thrombocytopenia; 1/26 hypothyroidism
Oudjik et al. [24]	AFL: 8/10(80%) SVT: 6/10 (60%) H: 62.5% NH: 75%	Maternal: 2/21 minor side effects (nausea, vomiting, dizziness)
Vigneswaran et al. [26]	Overall: 25/32 (78%) H: 12/14 (85.7%) NH: 13/18 (72.2%)	Not reported
Jaeggi et al. [18]		Maternal: nausea, dizziness attributed to digoxin (38%), flecainide (20%), sotalol (10%), visual disturbances with flecainide (14%); 1/111 electrolyte disturbance (combination therapy of flecainide and digoxin); 1/111 sotalol-induced bradycardia Fetal: bradycardia due to combination therapy of flecainide and sotalol
Sridharam et al. [21]	Short VA SVT: - Digoxin: 29/42 (69%) - Flecainide: 24/25 (96%) Long VA SVT: - Digoxin: 4/8 (50%) - Flecainide: 8/9 (88.9%) H: - Digoxin: 9/21 (38.1%) - Flecainide: 7/7 (100%) NH: - Digoxin: 23/29 (79%) - Flecainide: 26/27 (96.0%)	Flecainide: 8/34 (24%)- lightheadedness, nausea, headache, transient blurred vision, heightened alertness in mothers Digoxin: 2/50 maternal intolerance (1 case with psychiatric illness)
Strizek et al. [28]	Flecainide monotherapy H: 13/18 (72%) NH: 9/10 (90%)	Maternal: ECG with Brugada pattern, with spontaneous resolution after cessation of flecainide
Van Der Heijden et al. [25]	SVT: 14/18 (78%) AF: 8/10 (80%) H: 4/6 (67%) NH: 18/22 (81.8%)	Maternal: 15/28 minor symptoms (dizziness, fatigue, nausea, vomiting, headache)
Shah et al. [29]	AF: 5/5 (100%) SVT: 6/16 (37.5%)	Maternal: 4/21 minor side effects (nausea, dizziness, fatigue)
Miyoshi et al. [23]	Overall: 44/49 (89.9%) H: 3/4 (75%)	Maternal: 9/50 (78.0%) minor symptoms (most common: nausea, vomiting); 1/50 transient Mobitz II type AVblock Fetal: Fetal period: 1/50 severe bradycardia, 1/50-de novo arrhythmia Neonatal period: 1/50 ileus, 1/50 hypoglycemia
Ekiz et al. [19]	Flecainide monotherapy SVT: 15/17 (88.2%) H: 13/15 (86.6%)	Maternal: 1/23 atrial fibrillation with spontaneous resolution after cessation of flecainide; 1/23 dizziness
O’Leary et al. [30]	Overall conversion rate: 39/57 (68.4%) Total cases in which rhythm or rate control was achieved: 47/57 (83%)	Not reported
Broom et al. [31]	Overall treatment efficacy: - Flecainide: 6/9 (66.8%) - Digoxin: 10/28 (35.7%) - Sotalol: 3/6 (50%) - Digoxin+ Flecainide: 22/27 (81.5%) - Digoxin+ Sotalol: 7/13 (53.8%) - Overall: 48/56 (85.7%)	Maternal: 14/56 (25%) total side effects, from which 6/14 consisted of mild symptoms

H—hydropic patients; NH: non-hydropic patients; SVT—supraventricular tachycardia; AFL—atrial flutter;VT—ventricular tachycardia; JET—junctional ectopic tachycardia; VA—ventriculoatrial; AV—atrioventricular.

## Data Availability

The raw data presented in this study can be obtained upon reasonable request addressed to Liliana Gozar (lili_gozar@yahoo.com).

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
