# Peer review of "Fetal Tachyarrhythmia Management from Digoxin to Amiodarone—A Review"

_jcm, 2022, doi:10.3390/jcm11030804_

Round 1
Reviewer 1 Report
In this article, the authors performed a review of the literature to establish the optimal management of fetal tachyarrhythmias.
The authors carry out a narrative review of the literature. I believe that this topic, where controversy usually exists between three first-line treatments (digoxin, flecainide, and sotalol), can be better addressed by a systematic review of the literature. However, I also consider that an extensive and well-documented narrative review can be a good source of evidence where different nuances are appreciated, as long as it includes the entire body of evidence available in previous studies.
However, after reading the results, I think that many interesting related studies may have been omitted. There is a recent systematic review oriented to the study of transplacental treatment of fetal arrhythmias, which includes 21 studies (392 patients) (Hill GD et al. Prenat Diagn. 2017; PMID: 28833310). Another recent review of the literature studies a total of 537 patients with different treatment strategies (Alsaied T et al. J Am Heart Assoc. 2017; PMID: 29246961). As can be deduced from Tables 2 and 3, the manuscript finally includes results from 15 previous studies. Therefore, this review does not add new evidence to what was established in previous reviews, and omits the results of other interesting ones.
Furthermore, in the Results section, the authors do not indicate either the number of included studies, nor the number of fetuses that were finally studied. They do not indicate how many studies were reviewed, how many were excluded and the reason for exclusion. Even though it is not a systematic review of the literature, I believe that these aspects are critical for a correct assessment of the review carried out, and it gives coherence to the results.
The review should have included other databases such as Scopus or Web of Science in addition to PubMed, to include the entire body of evidence. This review also includes a very long time period (1980 - 2020), where fetal and pediatric cardiology has undergone a considerable change, and the authors interpret and compare the studies carried out in the 1990s with those published in the recent years. So I think the review should have been limited to studies done in the last 10-15 years, or at least use the review to demonstrate this change in management that this pathology has undergone in recent years.
Subsequently, the discussion is poorly structured, and is limited to summarizing the most relevant studies included in the review, without becoming a critical reading from which the authors draw relevant conclusions. Neither are clear recommendations based on the evidence obtained in the discussion. In the Conclusion section, no clear conclusions are issued either, and rather imprecise and incomplete management guidelines are reflected, without finally addressing such important aspects as the periodicity in the control (“more frequently”), the management of the fetus with hydrops, or really what would be the first-line drug for the authors as a result of the review carried out, limiting itself to stating that “flecainide should be considered”.
The authors conclude that, when establishing antiarrhythmic treatment, not only the fetus but also the mother must be taken into account before initiation, an aspect with which I agree. However, despite its importance, it is an aspect that is poorly addressed in the discussion. We recommend reading recent related articles (Chimenea Á et al. Eur J Obstet Gynecol Reprod Biol. 2021; PMID: 33276280).
Finally, I consider that the English grammar could be improved, and there are different correctable elements in the text, such as the presence of exclamations in the tables, which is unorthodox (see table 2).
In conclusion, I think it is an article with a lot of potential, but with an unambitious review of the literature, and that does not add new evidence to what has already been published. Furthermore, the discussion is not very focused, and management guidelines are not clearly established. For this reason, I consider that the article cannot be published in its current form.
Author Response
Dear Editor-in-Chief,
Please find attached for your consideration the revised version of the manuscript entitled “Fetal Tachyarrhythmia Management from Digoxin to Amiodarone- A Review”, written by Liliana Gozar, Dorottya Gabor-Miklosi, Rodica Toganel, Amalia Fagarasan, Horea Gozar, Daniela Toma and Andreea Cerghit Paler (Manuscript ID: jcm-1517295).
We wish to express our gratitude for all the comments and suggestions addressed to our manuscript, as they allowed us to improve our work significantly.
Within this revised manuscript we answered point-by-point all the addressed questions and suggestions by making the required modifications (marked in red color) following the reviewers’ observations as below:
Reviewer #1:
Comments and Suggestions for Authors
In this article, the authors performed a review of the literature to establish the optimal management of fetal tachyarrhythmias.
- The authors carry out a narrative review of the literature. I believe that this topic, where controversy usually exists between three first-line treatments (digoxin, flecainide, and sotalol), can be better addressed by a systematic review of the literature. However, I also consider that an extensive and well-documented narrative review can be a good source of evidence where different nuances are appreciated, as long as it includes the entire body of evidence available in previous studies.
Answer: We are grateful for your time and effort that you have dedicated to providing us with valuable feedback and comments on our manuscript. We try to write an extensive and well-documented narrative review and we thought that this it would give it robustness needed.
- However, after reading the results, I think that many interesting related studies may have been omitted. There is a recent systematic review oriented to the study of transplacental treatment of fetal arrhythmias, which includes 21 studies (392 patients) (Hill GD et al. Prenat Diagn. 2017; PMID: 28833310). Another recent review of the literature studies a total of 537 patients with different treatment strategies (Alsaied T et al. J Am Heart Assoc. 2017; PMID: 29246961). As can be deduced from Tables 2 and 3, the manuscript finally includes results from 15 previous studies. Therefore, this review does not add new evidence to what was established in previous reviews, and omits the results of other interesting ones.
Answer: Thank you for your suggestion. Both of the reviews that you mentioned are part of our research. The Alsaied T et al. and Hill GD et al. literature reviews are very interesting and related to our research. Those are our bibliographic reference number 1 and 23. In Tables 2 and 3, we included only original research studies.
It is true that 15 studies met the inclusion criteria; we excluded the case controls and the studies with insufficient data (for example the number of patients with hydrops, the adverse events present and the tachycardia type).
Although our review might seemingly not add new evidence to the field, its aim is to raise awareness regarding the changes that have occurred in the treatment of fetal tachycardia during the past years and summarize the experiences gathered. As digoxin still remains in the clinical practice in several institutions, despite evidence regarding its decadency. It is important to emphasize the changes that have occurred and encourage clinicians to adopt the new treatment protocols.
- Furthermore, in the Results section, the authors do not indicate either the number of included studies, nor the number of fetuses that were finally studied. They do not indicate how many studies were reviewed, how many were excluded and the reason for exclusion. Even though it is not a systematic review of the literature, I believe that these aspects are critical for a correct assessment of the review carried out, and it gives coherence to the results.
Answer: Thank you for this comment. We completed our Results section with your suggestion, with the following sentence „Fifteen studies met inclusion criteria, with 839 patients." (please see line 94). The inclusion and exclusion criteria are noted at lines 83-86.
- The review should have included other databases such as Scopus or Web of Science in addition to PubMed, to include the entire body of evidence. This review also includes a very long time period (1980 - 2020), where fetal and pediatric cardiology has undergone a considerable change, and the authors interpret and compare the studies carried out in the 1990s with those published in the recent years. So I think the review should have been limited to studies done in the last 10-15 years, or at least use the review to demonstrate this change in management that this pathology has undergone in recent years.
Answer: We included only Pubmed and Google Scholar databases because there we found sufficient and important studies related to our review and we did not consider that we needed another database. However, thank you for your suggestion.
It is true that the studies included in our review cover a large period, from 1980 to 2020, where fetal and pediatric cardiology has undergone a considerable change, thus great developments in the field could be observed through our review.
- Subsequently, the discussion is poorly structured, and is limited to summarizing the most relevant studies included in the review, without becoming a critical reading from which the authors draw relevant conclusions. Neither are clear recommendations based on the evidence obtained in the discussion. In the Conclusion section, no clear conclusions are issued either, and rather imprecise and incomplete management guidelines are reflected, without finally addressing such important aspects as the periodicity in the control (“more frequently”), the management of the fetus with hydrops, or really what would be the first-line drug for the authors as a result of the review carried out, limiting itself to stating that “flecainide should be considered”.
Answer: According to your suggestions, we have improved the discussion and conclusion sections, in an attempt to making them clearer and easier to understand.
- The authors conclude that, when establishing antiarrhythmic treatment, not only the fetus but also the mother must be taken into account before initiation, an aspect with which I agree. However, despite its importance, it is an aspect that is poorly addressed in the discussion. We recommend reading recent related articles (Chimenea Á et al. Eur J Obstet Gynecol Reprod Biol. 2021; PMID: 33276280).
Answer: Thank you for your suggestion. We added in the discussion section a subsection entitled "4.4. Maternal side effects", where we addressed the side effects of antiarrhythmic treatment for the mothers. (please see lines 230-256). Moreover, we included the article that you recommended
(Chimenea Á et al. Eur J Obstet Gynecol Reprod Biol. 2021; PMID: 33276280), please see lines 238-240.
- Finally, I consider that the English grammar could be improved, and there are different correctable elements in the text, such as the presence of exclamations in the tables, which is unorthodox (see table 2).
Answer: Regarding the language problem, our colleague reviewed this article for English language. We attach the certificate of professional competence. Moreover, we improved our tables.
Thus, by this letter and by the attached revised format of our manuscript we hope to have fulfilled all the observations and recommendations made by the Reviewer 1.
Thank you for your time and consideration.
On behalf of all authors of this work,
Yours sincerely,
Dorottya Gabor-Miklosi

Reviewer 2 Report
Thank you for allowing me to review the manuscript titled "Fetal Tachyarrhythmia Management from Digoxin to 2
Amiodarone- A Review " by Gozar et al.
Comments:
I have to applaud the authors for this important work in fetal arrhythmia management and including more than 40 articles that they reviewed for inclusion in this study.
The study is well written and I just have one question to ask the authors
Why not do a meta analysis instead of just stopping with a review to make it more robust?
Author Response
Dear Editor-in-Chief,
Please find attached for your consideration the revised version of the manuscript entitled “Fetal Tachyarrhythmia Management from Digoxin to Amiodarone- A Review”, written by Liliana Gozar, Dorottya Gabor-Miklosi, Rodica Toganel, Amalia Fagarasan, Horea Gozar, Daniela Toma and Andreea Cerghit Paler (Manuscript ID: jcm-1517295).
We wish to express our gratitude for all the comments and suggestions addressed to our manuscript, as they allowed us to improve our work significantly.
Within this revised manuscript we answered point-by-point all the addressed questions and suggestions by making the required modifications (marked in red color) following the reviewers’ observations as below:
Reviewer #2:
Comments and Suggestions for Authors
I have to applaud the authors for this important work in fetal arrhythmia management and including more than 40 articles that they reviewed for inclusion in this study.
The study is well written and I just have one question to ask the authors
- Why not do a meta analysis instead of just stopping with a review to make it more robust?
Answer: We are grateful for your time and effort that you have dedicated to providing us with valuable feedback on our manuscript. We try to write an extensive and well-documented narrative review and we thought that this it would give it robustness needed.
Thus, by this letter and by the attached revised format of our manuscript we hope that we answered the question made by the Reviewer 2.
Thank you for your time and consideration.
On behalf of all authors of this work,
Yours sincerely,
Dorottya Gabor-Miklosi

Round 2
Reviewer 1 Report
After the review, I consider that the authors have made a significant effort to improve the final result of the manuscript, and have taken into account the considerations made in the first review. Therefore, I believe the article is publishable in its current form.